# Parallel reverse genetic screening in mutant human cells using transcriptomics

Bianca V Gapp[1,†], Tomasz Konopka[1,†], Thomas Penz[2], Vineet Dalal[1], Tilmann Bürckstümmer[3], Christoph Bock[2,4,5] & Sebastian MB Nijman[1,2,6,*]

## Abstract

Reverse genetic screens have driven gene annotation and target discovery in model organisms. However, many disease-relevant genotypes and phenotypes cannot be studied in lower organisms. It is therefore essential to overcome technical hurdles associated with large-scale reverse genetics in human cells. Here, we establish a reverse genetic approach based on highly robust and sensitive multiplexed RNA sequencing of mutant human cells. We conduct 10 parallel screens using a collection of engineered haploid isogenic cell lines with knockouts covering tyrosine kinases and identify known and unexpected effects on signaling pathways. Our study provides proof of concept for a scalable approach to link genotype to phenotype in human cells, which has broad applications. In particular, it clears the way for systematic phenotyping of still poorly characterized human genes and for systematic study of uncharacterized genomic features associated with human disease.

**Keywords** kinases; multiplexed RNA sequencing; parallel screening; reverse genetics; systematic phenotyping
**Subject Categories** Chromatin, Epigenetics, Genomics & Functional Genomics; Methods & Resources
**Mol Syst Biol.** (2016) 12: 879

## Introduction

Forward and reverse genetic approaches have both been crucial for elucidating fundamental biological processes as well as identifying therapeutic targets. These approaches identify genes underlying a particular trait (forward genetics) or uncover phenotypes of particular mutants such as gene knockouts (reverse genetics). Forward genetic screening has been employed extensively in human cells using RNAi, gene trap, and CRISPR/Cas9 approaches (Lehner, 2013; Mohr *et al*, 2014; Shalem *et al*, 2015). In contrast, large-scale reverse genetic approaches in human cells have been limited to arrayed RNAi screens and typically only interrogated a single phenotype such as viability or changes in a particular signal transduction pathway (Brummelkamp *et al*, 2003; Paulsen *et al*, 2009; Zhang *et al*, 2009; Kranz & Boutros, 2014; Tiwana *et al*, 2015). Thus, deep phenotyping of gene mutants has been largely restricted to model organisms (Giaever *et al*, 2002; White *et al*, 2013; Shah *et al*, 2015).

One of the hurdles associated with large-scale reverse genetics in human cells is the technical challenge to generate large sets of individual, targeted mutants. Earlier methods such as RNAi provided a scalable method but suffer from incomplete knockdown and off-target effects that introduce substantial noise and hinder the interpretation of results (Kaelin, 2012). A second hurdle includes the comprehensive phenotyping of large sets of samples: Mammalian cells can contain thousands of features of potential interest and many of these are cell type specific. The net impact of these difficulties is the limitation of reverse genetic approaches in human cells to a small number of mutants. This slows down the study of fundamental human biology and hinders understanding of diseases. As many mutations are species specific, they cannot be modeled in other organisms. There is thus a need for a general, scalable, and accessible method for reverse genetics in human cells.

In this work, we exploit advances in parallel sequencing and genome editing (van Dijk *et al*, 2014; Barrangou *et al*, 2015) to revisit reverse genetics in human cells. We first establish a phenotypic profiling method based on RNA sequencing that is scalable and suitable for large-scale screening. We then perform 10 parallel screens in a collection of 64 mutant cell lines derived from a haploid parental line (Carette *et al*, 2010). The collection includes cells deficient in 55 individual tyrosine kinases.

## Results

Transcriptional profiling has been demonstrated in yeast to connect genotypes to phenotypes and is thus a suitable assay for reverse

1 Nuffield Department of Clinical Medicine, Ludwig Cancer Research Ltd., University of Oxford, Oxford, UK
2 CeMM Research Center for Molecular Medicine of the Austrian Academy of Sciences, Vienna, Austria
3 Horizon Genomics, Vienna, Austria
4 Department of Laboratory Medicine, Medical University of Vienna, Vienna, Austria
5 Max Planck Institute for Informatics, Saarbrücken, Germany
6 Nuffield Department of Clinical Medicine, Target Discovery Institute, University of Oxford, Oxford, UK
*Corresponding author. Tel: +44 1865 612885; E-mail: Sebastian.nijman@ludwig.ox.ac.uk
†These authors contributed equally to this work

genetics (DeRisi *et al*, 1997; Hughes *et al*, 2000). In particular, specific genetic, chemical, and environmental perturbations have been shown to yield gene expression signatures that provide insight into gene function (Holstege *et al*, 1998; Chua *et al*, 2006; Lamb *et al*, 2006; Hu *et al*, 2007; van Wageningen *et al*, 2010; Lenstra *et al*, 2011; Kemmeren *et al*, 2014). We wished to apply a similar strategy based on perturbations to study human cells. We reasoned that shallow sequencing of mRNA, previously deployed to measure single-cell transcriptomes (Wu *et al*, 2014), would provide the throughput required for screening applications while maintaining sufficient resolution to capture expression changes. We thus decided to measure transcriptional profiles using a library preparation protocol that amplifies the 3′ ends of transcripts and is designed to facilitate multiplexing.

To explore and benchmark shallow sequencing for systematic screening, we performed perturbation experiments in human HAP1 cells (Carette *et al*, 2010). Cells were cultured under reduced serum conditions for 16 h and stimulated with seventy diverse stimuli, including polypeptides and small molecules (Fig EV1A and Table EV1). Most conditions were measured in two biological replicates, and 48 samples were combined per Illumina HiSeq lane, yielding 2–4 million reads per sample. Expression profiles of

replicate samples were strongly correlated, indicating robust and consistent performance of the assay (Fig 1A). Modeling of sequencing depth showed that measuring ~1 million reads per sample was sufficient to identify nearly all the ~12,000 genes expressed in HAP1 cells (Fig 1B). Moreover, we estimated that our depth range should enable us to call upregulation of expression by a twofold change in around two-thirds and upregulation with threefold change in more than 90% of the expressed genes.

Through comparison of stimulated to mock-treated samples, we determined sample-wise signatures of differentially expressed genes. We also computed group-wise signatures using concordance across replicate samples. Using data from a stimulation performed in eight replicates, we estimated that group-wise signatures were robust for screening when based on just two replicates (Fig EV1B). Together, these technical metrics indicate that the approach produces gene signatures that are informative.

Next, we studied the specific signatures induced by our panel of stimuli. Around half of the stimuli elicited discernible transcriptional responses of up to ~200 genes under the chosen experimental conditions. Absence of signatures for several of the stimuli could be due to timing, dosing, assay sensitivity, or true unresponsiveness. Gene ontology analysis of signature genes identified pathways previously

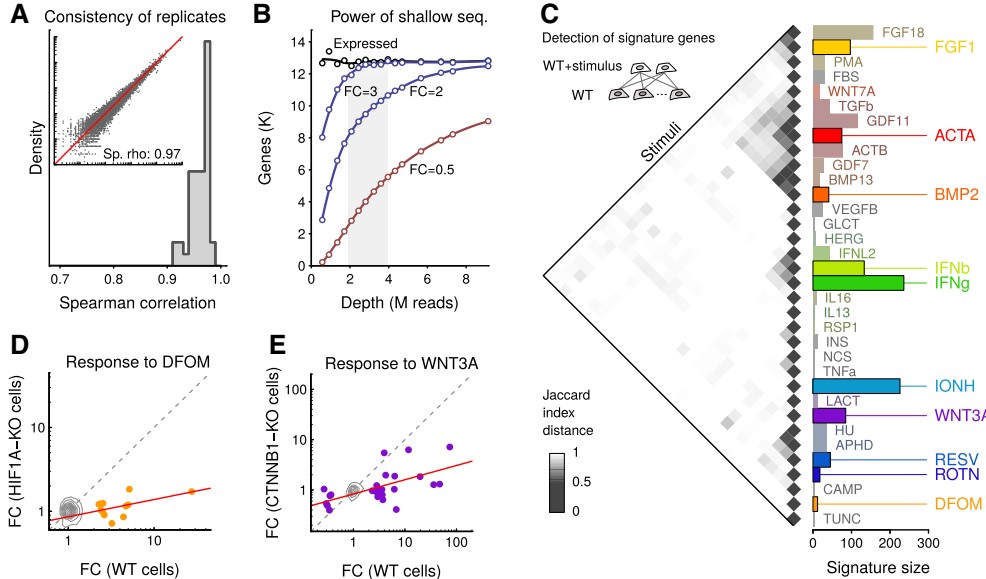

**Figure 1.** **A platform for large-scale cell profiling by shallow RNA sequencing.**

A Spearman correlations between replicates of expression profiles in HAP1 cells measured by shallow RNA-seq. Libraries were prepared using a protocol capturing 3-prime ends of polyadenylated transcripts. Inset shows gene expression values in a representative pair of replicates.

B Data-based modeling of the effect of sequencing depth on gene expression analysis. Dots represent synthetic samples obtained by pooling 24 HAP1 wild-type sequencing runs and subsampling. Line labeled "Expressed" shows the number of genes that can be detected with expression above a threshold (transcripts per million reads above 1). Lines labeled with FC show estimates of the number of genes that could be detected as differentially expressed were their expression to change by the indicated factor. FC, fold change; K, thousand; M, million.

C Clustering of signature gene sets from polypeptide and small molecule stimulations. Inset shows strategy for obtaining gene signatures wherein each stimulated sample is compared to a control set, and a signature is obtained by consensus of two replicates. The heatmap shows a clustering of stimuli wherein similarities are assessed by 1 − Jaccard index of the signature sets. The bar chart displays sizes of signature sets. Solid colors indicate a panel of diverse stimuli selected for the 10 reverse genetic screens. WT, wild type.

D Comparison of expression profiles of wild-type cells and HIF1A-KO cells in response to DFOM stimulation. Contours depict genes not differentially expressed; dots indicate DFOM signature genes; gray dotted line is the diagonal of equal response; and red line is a linear fit using signature genes. FC, fold change; WT, wild type; KO, knockout.

E Same as in (D), except for WNT3A stimulus in CTNNB1-KO cells.

linked with the tested stimuli (Table EV2). Signatures for related stimuli clustered together (Figs 1C and EV1C). For example, members of the TGF-beta superfamily (TGFb, ACTA, GDF11, ACTB, BMP2, GDF7, BMP13) formed one large cluster. Interferon-beta (IFNb), interferon-lambda (IFNL2), and interferon-gamma (IFNg) formed a separate cluster. Importantly, although related signatures (e.g., interferons) contained genes in common, they also contained gene subsets known to be specific to the respective stimuli (Fig EV1D). This indicates that the resolution of shallow RNA sequencing can capture not only broad responses to perturbations, but can reveal nuances of signaling cascades as well.

Satisfactory performance of shallow transcriptomic profiling prompted us to carry out the first transcriptome-based reverse genetic screen in human cells. As many signaling pathways are inactive under standard culturing conditions, we reasoned that phenotypes associated with gene knockouts would only become apparent upon a secondary perturbation (Lamb *et al*, 2006; Kemmeren *et al*, 2014). We thus selected 10 stimuli from the benchmarking experiment based on signature size and diversity for parallel screening. These were activin A (ACTA), bone morphogenic protein 2 (BMP2), fibroblast growth factor 1 (FGF1), IFNb, IFNg, wingless-type family member 3A (WNT3A), deferoxamine (DFOM, hypoxia mimicking agent), rotenone (ROTN, inducer of reactive oxygen species), resveratrol (RESV, a natural product with unclear mode of action), and ionomycin (IONM, calcium modulating agent). To strengthen confidence in these selected gene signatures, we collected additional replicates under the same conditions, with the exception of ionomycin for which we lowered dosage due to cytotoxicity. The final signatures were consistent with our initial findings (Fig EV1E).

Next, we validated that the previously defined signatures can be exploited to functionally annotate genes using mutant cell lines. We selected a small (induced by DFOM) and a medium size signature (induced by WNT3A) and tested whether specific knockouts would affect these signatures. Using CRISPR/Cas9 genome editing, we generated HAP1 cells deficient for HIF1A or CTNNB1 (beta-catenin), critical and specific transcription factors in hypoxia and WNT signaling. As expected, genes upregulated by DFOM and WNT3A were strongly reduced in the HIF1A and CTNNB1 mutants, respectively (Fig 1D and E).

Finally, we tested whether we could also uncover genotype–phenotype connections in a large unbiased setting. We chose to focus on tyrosine kinases as these represent a recognized class of drug targets, yet many of the 90 members encoded in the human genome remain poorly annotated (Fedorov *et al*, 2010). Based on essentiality in HAP1 cells (Blomen *et al*, 2015) and RNA expression, we selected 56 tyrosine kinases, and for each gene, we attempted to generate isogenic knockout clones in HAP1 cells using CRISPR/Cas9 (Fig 2A and Appendix Fig S1). Guide RNAs were designed to target coding exons at least 100 bp downstream of the start codon to avoid translational initiation from a downstream ATG. Mutant clones were expanded, and gene knockout was confirmed by Sanger sequencing in more than 95% (55/56) of the selected genes. The great majority of clones was morphologically indistinguishable from wild-type cells and proliferated at similar speed.

We adopted a scalable and modular screen design, splitting data acquisition into batches. Each batch consisted of four

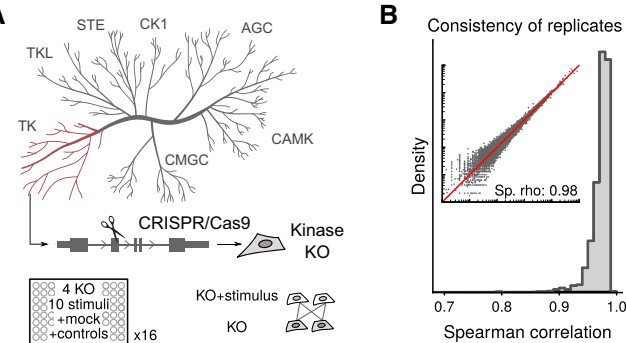

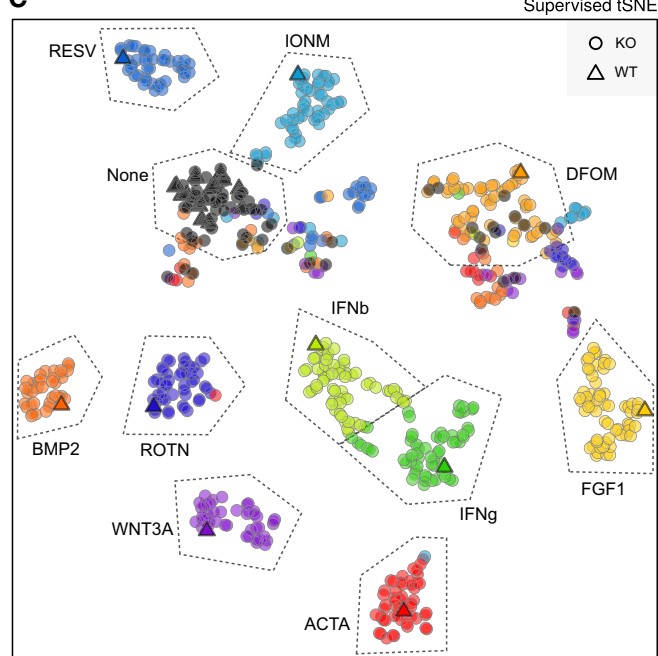

**Figure 2.  Parallel reverse genetic screening of kinase knockout cells.**

A   On top, cartoon illustrating the assembly of a collection of HAP1 knockouts using CRISPR/Cas9 technology. Abbreviations indicate kinase subfamilies. At bottom, scheme for screening design showing that individual kinase KO cells are measured along all relevant controls. KO, knockout.

B   Spearman correlations between replicates of stimulated and unstimulated wild-type and knockout cells in the transcriptomic screen of 16 96-well plates. Inset shows expression values in a representative set of replicates.

C   Supervised Stochastic Neighbour Embedding (tSNE) clustering of all stimulated and unstimulated HAP1 wild-type and knockout cell lines. Dots represent averages of replicates. WT, wild type; KO, knockout.

knockout cell lines screened in parallel against the 10 selected stimuli along with controls (Fig EV1F). This allowed us to maintain replicates and mutant-specific samples in one batch, reducing the need for batch correction for some analyses (see Materials and Methods). In this manner, we processed 64 HAP1 knockout cell lines (55 tyrosine, 6 nontyrosine kinases, and 3 positive controls) and again obtained high concordance in expression profiles between replicates (Fig 2B). Clustering based on the defined signatures showed the expected groupings by stimulus (Fig 2C), indicating that most mutant cell lines responded to the

perturbations similarly to wild-type cells. Interestingly, around 15% of the knockout cells showed signatures with substantial overlap with the DFOM signature (Appendix Fig S2A), explaining the imperfect clustering of DFOM samples and some unstimulated controls. Although this effect was less strong than that induced with DFOM, it suggests that some clones had an activated hypoxia response under normoxic conditions. Indeed, Western blot analysis showed that HIF1A protein levels were elevated under normoxic conditions in clones displaying the hypoxia-like signature (Appendix Fig S2B and C). The levels were comparable to those observed in DFOM-treated cells. However, this increase was not consistently observed in independently generated knockout clones, suggesting that the hypoxic state is a modestly frequent (~15%) passenger effect.

Further analysis of the screening data indicated that responses of mutant cells to the stimuli were weakly correlated with RNA concentration and sequencing depth (Appendix Fig S3). This suggested that cell growth, albeit largely managed experimentally, had a measurable effect on the signatures, highlighting the potential confounding effects of cell cycle and cell density on cellular responses. We thus created linear models to correct for these effects and used residuals to score individual cell lines' responses to each stimulus (Fig EV2). This revealed several knockout-specific signaling dependencies (Figs 3A–C and EV3). For example, JAK1 knockout cells were completely insensitive to IFNg and IFNb while responding similarly as wild-type cells to the other eight stimuli. In contrast, JAK2 or TYK2 ablation did not affect the response to interferon under these conditions (Figs 3B and EV4). This finding is surprising as these three JAK family members have been reported to contribute to a transcriptional response upon stimulation with type I or type II interferons (Rane & Reddy, 2000). Our results confirm a critical role for JAK1 in interferon signaling and suggest a distinct

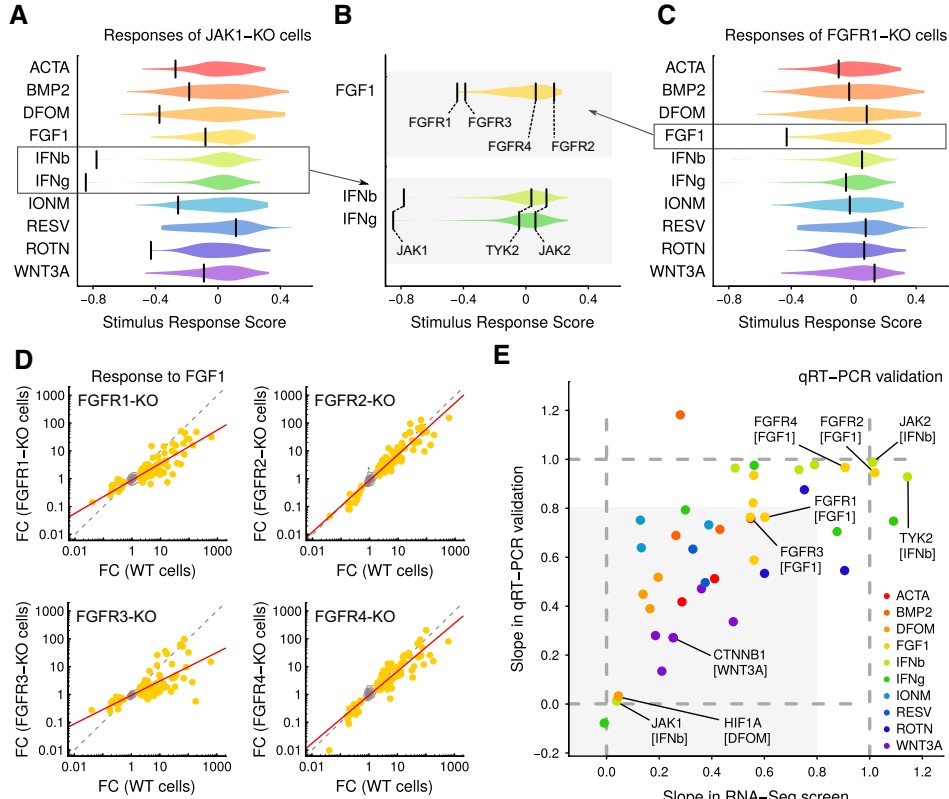

**Figure 3. Transcriptional profiling of kinase knockouts links genotypes to pathways.**

A  Responses of JAK1-KO cells to the ten selected stimuli. Violins indicate score distributions of all knockout cell lines. Scores are overlaps of signature gene sets with expected signature sets, corrected for technical variables using general linear models. Bars represent scores for JAK1-KO mutants. KO, knockout.

B  Similar as in Fig 3A, except showing detailed view of responses to FGF1 and IFNb/IFNg stimulation of selected knockout cells. Bars indicate labeled mutants of FGFR and JAK family members.

C  Same as in Fig 3A, except showing FGFR1-KO cell line.

D  Comparison of response signatures in wild-type and FGFR-KO mutant cells. Contours summarize genes that are not differentially expressed; dots indicate FGF1 signature genes; gray dotted line is the diagonal of equal response; and red line is a linear fit using signature genes. FC, fold change; WT, wild type; KO, knockout.

E  Comparison of stimulus response as measured by RNA-seq and qRT–PCR. Each axis shows the slope of a best-fit line through KO and WT stimulus responses (lines for RNA-seq are as in Fig 3D, and lines for qRT–PCR data are computed similarly from independent stimulation and qRT measurements). Dotted lines are guides representing unit slope (equal response in KO and WT cells to stimulus) and zero slope (KO cells fully unresponsive to stimulus). Shaded area represents the space where both assays indicate that KO cells are less responsive than WT cells. WT, wild type; KO, knockout.

function of this kinase compared to the other two family members, at least in HAP1 cells.

As another example, we noted differential responsiveness of mutants in the FGF receptor (FGFR) family, which bind FGF1. Signaling through these receptors occurs through overlapping downstream cascades (Raju *et al*, 2014), but is also context dependent. Accordingly, mutations in distinct FGFR family members are associated with specific cancers (Touat *et al*, 2015). In HAP1 cells, the response to FGF1 was diminished through knockout of FGFR1 and FGFR3, but not FGFR2 or FGFR4 (Fig 3B and C). Studying the signature genes in more detail, we further noted that loss of FGFR1 had a uniform effect on FGF1 signaling, as marked by an overall reduction in the strength of the response (Figs 3D and EV4). In contrast, in FGFR3 knockout cells, the attenuation was less uniform. These observations highlight the complexity of FGF1 signaling and illustrate how the profiling platform can spark new hypotheses even for well-studied pathways.

Many other gene–stimulus combinations also resulted in subtle reductions in signaling strength. To assess whether these small effects were reproducible, we selected gene–stimulus combinations across all the stimuli and validated them using qRT–PCR (Fig 3E). Remarkably, changes in stimulus response were quantitatively consistent with the results seen in the screens. These experiments also confirmed another observation that some mutant clones show aberrations in more than one signaling pathway (Fig EV5).

## Discussion

In summary, we present an approach for parallel reverse genetics of mutant human cells based on shallow RNA sequencing. Besides demonstrating its suitability for studying cellular perturbations, we generated a proof-of-concept dataset comprising 11 conditions in a collection of 64 isogenic haploid mutant cell lines. This represents one of the largest transcriptomic experiments performed in a single cell line and demonstrates the scalability and suitability of the approach for exploring signaling mechanisms in human cells in a systematic manner.

There are some potential limitations of the genetic screening strategy. The resolution of shallow RNA-seq is not as high as obtained from deeper sequencing protocols. Changes in lowly expressed genes may thus be missed, but this loss is offset by the reduced cost of the assay that allows analysis of a higher number of samples. Furthermore, cellular changes that do not affect gene transcription, or only very transiently, cannot be quantified using this method. The generation of full knockout mutants in diploid cells may be less efficient than reported here, and we do not formally demonstrate that the shallow RNA-seq performs equally well on other (diploid) mammalian cell systems. Nonetheless, we anticipate that the strategy of transcriptional screening of mutant cells is generic and can be applied to study many other cellular systems provided relevant reference/control signatures are measured. Furthermore, the presented strategy can be deployed to address a multitude of biological questions beyond the study of full knockout mutants. Envisioned applications include hit validation and targeted hypothesis testing that are difficult to tackle through forward genetics.

## Materials and Methods

### Cell lines

Cells were propagated in Iscove's modified Dulbecco's medium (IMDM+GlutaMAX, Invitrogen GIBCO) supplemented with 10% heat-inactivated bovine serum (FBS, Invitrogen GIBCO), 100 μg/ml penicillin, and 100 μg/ml streptomycin (Sigma-Aldrich). All cell lines were grown at 37°C in a 5% $CO_2$-humidified incubator. HAP1 knockout cell lines were generated at Horizon Genomics (Table EV3).

A set of nonessential and expressed kinases was obtained by intersecting published datasets of human kinases (Manning *et al*, 2002), expressed genes in HAP1 cells (Essletzbichler *et al*, 2014), and nonessential genes in HAP1 cells (Blomen *et al*, 2015). Guide RNAs (gRNA) were designed to target coding exons of the genes of interest, preferentially targeting within the first 25% of the coding sequence and at least 100 bp downstream of the start codon to avoid translational initiation from a downstream ATG. Specificity of each gRNA was assessed using the Broad algorithm (http://crispr. mit.edu/). Cloning was performed by ligating oligonucleotides containing the gRNA sequence and the chimeric gRNA backbone into a plasmid harboring the U6 promoter.

To generate HAP1 mutants for screening, cells were transfected with expression plasmids encoding *Streptococcus pyogenes* Cas9 (pX165 from the Zhang lab), a gRNA, and a blasticidin resistance gene using TurboFectin (Origene). Untransfected cells were eliminated by treating HAP1 cells with 20 μg/ml blasticidin for 24 h. Cells were allowed to recover from antibiotic selection for 5–7 days, and clonal cell lines were isolated by limiting dilution. DNA was isolated from cells using the Direct PCR-Cell Kit (PeqLab). The region around the gRNA target site was amplified by PCR, and PCR products were analyzed by Sanger sequencing. Clones bearing frameshift mutations were selected and stored for use. Cells lines are available through Horizon Genomics.

Independently generated FGFR3 and PDGFRA knockout cell lines were obtained by ligating oligonucleotides encoding for the gRNA sequence (FGFR3: CAGCAGGAGCAGTTGGTCTT; PDGFRA: GCG TTCCTGGTCTTAGGCTG) with a lentiCRISPR v2 vector (Addgene #52961). Following lentiviral transduction, infected cells were selected with 0.5 μg/ml puromycin for 3 days. Clonal cell lines were isolated by limiting dilution and gDNA isolated using DNeasy Blood & Tissue kit (Qiagen) according to the manufacturer's instructions. Regions flanking the gRNA target site were amplified by PCR and analyzed by Sanger sequencing. Clones harboring frameshift mutations were expanded for follow-up experiments.

### Reagents and stimulation of cells

Recombinant polypeptides and small molecules were purchased from different vendors (Table EV1). Polypeptides were diluted in water, 0.1% BSA, 0.1% acetic acid, 10 mM sodium citrate (pH 3), 5 mM sodium phosphate (pH 8 or 7.2), or 10 mM acetic acid. Stocks were prepared in PBS containing 0.1% BSA. Small molecules were diluted in water, DMSO, or 20 mM MES buffer (pH 5.5).

Stimulation experiments were carried out in a 12-well format using $2 \times 10^5$ cells per well. Thirty-six hours after seeding, cells were washed twice with PBS, and IMDM supplemented with 0.5%

FBS, 100 μg/ml penicillin and 100 μg/ml streptomycin was added. After 16 h reduced serum conditions, cells were stimulated with polypeptides or small molecules for 6 h. Samples were washed twice with 1 ml PBS (pre-chilled to 4°C) and immediately stored at −80°C.

## RNA sequencing

Total RNA was isolated using RNeasy Mini kit (Qiagen) according to the manufacturer's instructions. 500 ng total RNA was used for library preparation using the QuantSeq 3′ mRNA-Seq Library Prep Kit (Lexogen) according to the manufacturer's protocol with the exception of using 13 instead of 12 PCR cycles for library amplification. Library concentrations were measured using Qubit dsDNA HS assay on a Qubit 2.0 Fluorometric Quantitation System (Life Technologies). Size distribution of pooled final libraries (48 samples) was assessed using Experion DNA 1K analysis kit on an Experion automated electrophoresis system (Bio-Rad). Libraries were diluted, and the T-fill reaction was performed on a cBot as described previously (Wilkening *et al*, 2013) with the exception that the T-fill solution was provided in a primer tube strip. For cluster generation, the cBot protocol SR Amp Lin Block TubeStripHyp v8.0.xml was used. Sequencing was performed on an Illumina HiSeq 2000 machine using 50-bp single-read v3 chemistry.

## Quantitative real-time PCR

Total RNA was isolated using RNeasy Mini kit (Qiagen), and DNase digest was performed using a TURBO DNase kit (Ambion) according to the manufacturer's protocols. About 500 ng to 1 μg total RNA was reverse-transcribed using random hexamer primers and RevertAid Reverse Transcriptase kit (Fermentas). cDNA synthesis was carried out according to the manufacturer's instructions (synthesis cycle: 10 min at 25°C, 60 min at 42°C, and 10 min at 70°C). About 25–50 ng of cDNA and 500 nM forward and reverse primer were used for PCR amplification with KAPA ABI Prism SYBR Fast (Kapa Biosystems) according to the manufacturer's instructions (synthesis cycle: 3 min at 95°C and (3 s at 95°C, 30 s at 60°C) × 40). Primers used for qRT–PCR are listed in Table EV4.

## Western blotting

Whole-cell lysates were prepared using 4× sample buffer (320 mM Tris–HCl pH 6.8, 40% glycerol, 16 μg/ml bromophenol blue, 8% SDS) containing 10% 2-mercaptoethanol (Fisher Scientific), incubated for 10 min at 95°C and subjected to SDS–PAGE (NuPAGE 4–12% Bis-Tris Gel, Invitrogen). Proteins were separated for 1.5 h at 130 V and transferred to a polyvinylidene difluoride (PVDF, Amersham Hybond-P, GE Healthcare) membrane for 2 h at 400 mA. Membranes were blocked with 0.2% Tropix I-Block (Applied Biosystems) for 1 h and incubated with primary antibody diluted in 0.2% Tropix I-Block overnight at 4°C. Primary antibodies and dilutions used were as follows: mouse anti-HIF1A (1:2,000) from BD Biosciences (610959) and rabbit anti-actin (1:1,000) from Sigma-Aldrich (A2066). Blots were washed with PBS containing 0.1% Tween-20 and incubated with

HRP-conjugated secondary antibodies (anti-mouse or anti-rabbit IgG from Bio-Rad diluted 1:10,000 in 0.2% Tropix I-Block) for 1 h at room temperature. HRP was detected using Western Lightning Plus-ECL (PerkinElmer).

## RNA-seq data processing and alignment

Unaligned reads in fastq format were trimmed of adapter sequence AGATCGGAAGAGCACACGTCTGAACTCCAGTCAC using Cutadapt (v.1.2.1) and then partitioned using TriageTools (Fimereli *et al*, 2013) (v0.2.2) to select long (–length 35), high-quality (–quality 9), and sequence-complex (–lzw 0.33) reads. Selected reads were aligned using GSNAP (Wu & Nacu, 2010; v2014-02-28) onto a custom genome index (gmap_build -k 14 -q 2) based on hg19 supplemented with ERCC92 transcript sequences. Expression estimates on Gencode V19 genes were collected from the alignments using Exp3p. This procedure implements read counting on gene bodies and normalizes by total sequencing depth; because the RNA sequencing protocol is designed to capture one read per transcript through the polyadenylated tail, expression normalization does not include the length of the gene body.

## Expression analysis

Analysis was performed in a series of modules built around a custom toolkit, ExpCube. The analysis was split into two parts. The first part consisted of analysis of four 96-well plates representing the stimulus discovery phase of the project. The second part was an extension to the entire dataset (twenty 96-well plates). Analysis modules and their dependencies are illustrated in Appendix Fig S4.

We began by gathering expression data from all samples into one object. This object included central estimates as well as intervals for each gene in each sample. Common steps in expression analysis are normalization and batch correction. However, by examining profiles of unstimulated wild-type HAP1 cells and controls, we observed that various implementations of these steps highlighted parts of the signal and hid others, making it difficult to select a unified scheme for the entire screen. Furthermore, the experimental design was such that most intended comparisons were between samples within a single batch, mitigating the need for explicit batch correction. For these reasons, we chose not to adjust the central expression values. Instead, we used within-plate and across-plate variation for unstimulated wild-type samples (of which there are four or more replicates per plate) to adjust uncertainty intervals. For each gene, we computed quantiles among replicates in each plate and quantiles among group averages across plates. We then compared this empirical variability to the base Poisson intervals and obtained a rescaling factor for each gene's Poisson interval. For 98% of genes, the across-plate variability was larger than the within-plate variability (median ratio equal to 1.8), indicating the importance of replicates of wild-type controls in each of the library preparation plates. We applied this interval rescaling operation to all samples in the screen. Thus, we incorporated empirical data on reproducibility of comparable samples from across the screen into the expression profiles of all other samples.

We scored differential expression (DE) based on effect sizes (fold changes) and uncertainty levels (z-scores, defined as differences in

central expression values divided by a joint estimate of interval size). Outlier samples due to failed sequencing were excluded from the analysis. Considering groups A and B, we scored each sample in group A against each sample in group B, one gene at a time. We set a score of + 1 for a *z*-score > 1.75 and a fold change > 1.75, a score of 0 for a *z*-score < 1.25 and a fold change < 1.25, and a linear gradient of scores for intermediate cases (negative values for downregulation). Through the *z*-score component, this approach penalized inconsistent/unreliable genes whose intervals were substantially modified in the previous step. We then obtained a group-level DE score using the mean of the sample-level scores. By construction, these scores lie in [−1, 1] and carry the same interpretation independently of the number of samples per group, albeit with some variability with very few replicates (Fig EV1B). We declared a gene to be in a signature if the score was > 0.7.

For the power analysis, we began by pooling raw data from 24 replicates of unstimulated wild-type cells from the stimulus discovery phase. We then subset the pool into bins of varying size and applied our alignment and expression-calling pipeline on each bin. From these expression profiles, we computed the number of genes with expression above 1 transcript per million reads. We also created hypothetical profiles with genes over- or under-expressed by various fold changes and applied our criteria to call differential expression. The number of genes called in this analysis reflects the sensitivity of the method to identify expression changes under uncertainty due to low-coverage and biological variability. This calculation is presented in the ExpCube package vignette.

For stimulus selection in the discovery phase, we worked with stimuli whose group signature contained at least two replicates and at least two signature genes. Clustering of stimuli was performed using a Jaccard index distance between signature gene sets. Gene set enrichment analysis was performed by comparing signature genes with a background set of expressed genes in HAP1 cells using the topGO package (Alexa *et al*, 2006).

In the screening phase, we compared overlaps for each stimulus and each mutant cell line with the expected responses in wild-type cells. We collapsed expression profiles onto the ten selected signatures and then performed tSNE (van der Maaten & Hinton, 2008) clustering based on Euclidean distances between groups using the dimensionally reduced data. For more detailed analysis, we correlated overlaps with technical features and noted unintentional relations with RNA concentration and depth (Appendix Fig S3). To correct for these effects, we set up general linear models (GLM) of the form O = *a*R + *b*D, where O denotes overlap, R is average RNA concentration (ng per ul), D is average sequencing depth (millions of reads), and *a* and *b* are coefficients. We then defined a stimulus response score as the residuals between observed and modeled overlap. Extreme values of this score identify outlying cell lines, that is, mutants showing abnormal response given cell density and sequencing performance.

For comparison between RNA-seq and qRT–PCR data, we computed slopes of best-fit lines between KO and WT responses plotted on logarithmic scales. Linear fits on log axes suggest a model where KO response is a power of the WT response, but we do not mean to emphasize this interpretation. Rather, we regard the linear fit as a convenient summary of the overall patterns with few fitted parameters. In the case of RNA-seq data, the best-fit line was computed using signature genes with one outlier removed. In the case of qRT–PCR, the line was fit using four signature genes and GAPDH.

## Data availability

All raw sequencing data have been deposited in the European Nucleotide Archive under accession ERP012914. Exp3p software is available at https://github.com/tkonopka/Exp3p (v0.1). ExpCube software is available at https://github.com/tkonopka/ExpCube. Additional code, data files, and processed expression values are available at https://zenodo.org/record/51842.

**Expanded View** for this article is available online.

## Acknowledgements

We would like to thank the Biomedical Sequencing Facility at CeMM for carrying out RNA sequencing using a custom T-fill protocol and Michael Schuster for quality control and initial processing of the sequencing data. We thank Michel Owusu for technical assistance. We also wish to acknowledge the Computational Biology Research Group Oxford for use of their services in this project. We thank Toolgen for their contribution to the kinase knockout collection and Lexogen GmbH for RNA sequencing protocol development. We thank Helen Pickersgill of Life Science Editors and Mary Muers for critical reading and editing of the manuscript. The research leading to these results has received funding from the European Research Council under the European Union's Seventh Framework Programme (FP7/2007-2013)/ERC grant agreement no. [311166]. B. V. G. is supported by a Boehringer Ingelheim Fonds PhD fellowship.

## Author contributions

BVG designed, executed, and interpreted benchmarking experiments, reverse genetic screening, and validation experiments. TK designed, analyzed, and interpreted benchmarking experiments, screening data, and validation experiments. TP performed T-fill reactions and RNA-seq. VD performed qRT–PCR and Western blot validation experiments. TB generated kinase knockout cell lines. CB supervised RNA-seq experiments and provided overall guidance. SMBN designed and interpreted experiments, directed the study, and provided overall guidance. BVG, TK, and SMBN assembled figures and wrote the manuscript.

## Conflict of interest

SMBN is a co-founder and shareholder of Haplogen GmbH. The company employs haploid genetics in the area of infectious disease. TB is an employee of Horizon Genomics GmbH. The company generated the human tyrosine kinase knockout collection based on HAP1 cells.

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
