## [Review Process File · Molecular Systems Biology]

Parallel reverse genetic screening in mutant human cells using transcriptomics

Bianca V. Gapp, Tomasz Konopka, Thomas Penz, Vineet Dalal, Tilmann Bürckstümmer, Christoph Bock and Sebastian M. B. Nijman

Corresponding author: Sebastian M. B. Nijman, University of Oxford

Review timeline:	Submission date:	16 February 2016
	Editorial Decision:	04 April 2016
	Revision received:	23 May 2016
	Editorial Decision:	01 July 2016
	Revision received:	06 July 2016
	Accepted:	07 July 2016

Maria Polychronidou

Transaction Report:

1st Editorial Decision

04 April 2016

Thank you again for submitting your work to Molecular Systems Biology. We have now heard back from the four referees who agreed to evaluate your study. As you will see below, the reviewers think that the proposed approach seems interesting and is likely to be relevant for the scientific community. However, they raise a number of concerns, which should be carefully addressed in a revision of the manuscript. The reviewers' recommendations are quite clear so there is no need to repeat the points listed below.

REFeree COMMENTS

Reviewer #1:

Referee report Gapp et al., "Parallel reverse genetic screening....", MSB

Summary and major points of critique

The study by Gapp and co-workers is an expression-profiling analysis of a collection of CRISPR-Cas9 kinase knock-outs in human HAP1 (haploid) cells, under a set of 10 stimuli. This combination (the collection of knock-outs, the expression-profiling and the fact that the experiments have been carried out under a set of different conditions) would be expected to make an exciting study. However, as it stands, in essence this does not go beyond a methods paper and one whereby the methodology is not conceptually new. Furthermore, the paper lacks a clearly new interesting biological finding (either general to all kinases or specific to a particular pathway), and certainly no

interesting finding that is backed up by appropriate follow-up analyses/experiments. There are major issues with the way the data has been presented and the way that some of the steps/analyses have been carried out. Although the approach would seem to be exciting (based on previous studies in model organisms) this is not really exemplified in this study which altogether leaves an impression of superficiality, in particular because of the lack of an interesting new biological insight. Besides these major issues there are many minor issues regarding the presentation/write-up.

Individual comments

In the introduction and in the main text ("first reverse genetics screens"), the authors state that reverse genetics has been limited to model organisms. This is incorrect for several reasons. Small-scale reverse genetics (one gene) in human cells has been going on for some time. Large-scale reverse genetics (many genes) has also been around for some time in the form of RNAi library screening approaches. There are obviously caveats to the latter that CRISPR-Cas9 will overcome, but such statements are incorrect. The (extremely short) introduction would benefit from a lot more subtlety.

(With regard to the hurdles alluded to in the introduction the authors could add a third - cell type specific phenotyping.)

The authors state that 300 knock-out clones have been generated for protein kinases with 90% of selected genes resulting in expandable clones. This superficiality is a pity and a missed opportunity. How many protein kinases are there in total? Did the authors target all? If not, what were the selection criteria? If they targeted all, then they should say something about essentiality (how many and which) discussing this in light of the systematically generated data of protein kinases from other organisms for which this is known (eg yeast). Furthermore, is the 90% a subset of the 300? If so how many exactly were made successfully (ie expandable: the 90% and 300 numbers come across as rounded-off numbers although they may be exact) and how many were then actually screened across the 10 conditions? It seems that although 300 kinase ko lines have been generated, only 64 have been profiled. This should be made more explicit (for example in the abstract) and the selection criteria needs to be described. It should also be noted that having only profiled a subset severely damages the otherwise potential systematic nature of this study. Initially doing all under a single condition and a subset under all 10 would already have been a step in the right direction.

The observation that some of the unstimulated ko's had a hypoxia-like response needs further work. Is this a result of the ko, or does it imply that a subset of unstimulated ko lines were inadvertently cultured under suboptimal conditions. If the former is true then this implies that these kinases all either have a role in suppressing such a response (probably unlikely if the number is large and includes several unrelated kinases). A third possibility is that this represents a frequently occurring indirect effect. Whatever the case the observation begs for a proper explanation.

Given the technological nature of the current study it would be a good idea to describe in more detail the extent of the correlations between apparent response and sequencing depth and RNA concentration. This should lead to a precise recommendation on how this can be avoided in future studies. Although linear modeling can certainly help to correct for such effects, future studies would benefit from having to avoid such corrections.

Typo: page 6 last word "is" can be removed

Figures 1A, 2A, 2C are cartoon representations of the experiment set-up. These are a complete waste of space and should be removed to the supplemental figures, leaving more space for presenting actual findings. None of the experimental strategies being employed are at all difficult to comprehend by reading the text and it is difficult to understand why the authors think that it's a good idea to fill almost one third of their figures with such cartoons rather than data.

All data being presented is derived through compilation of results on individual genes. While this is certainly a requirement for such studies, it would increase the confidence in the data if the authors also presented some of their results in the form of figures that show results on individual genes, for example with heatmaps. In this way things like similarity or differences can be better judged than through compiled results only.

The authors present reproducibility in the form of a correlation of ranking (Spearman correlation Fig 1B, 2C). This is in itself already a red flag since it is much easier to get similar ranking than similar values. In addition although the correlation is derived by a test of similarity of ranking the example scatterplot doesn't seem to be plotting the rank, rather the expression level. This is incorrect. Even worse, when demonstrating differences a different measure is used, not based on ranking. It is wholly incorrect to first demonstrate reproducibility by one measure and then analyze differences by another. Reproducibility should be tested by the same type of correlation used to assess differences between different samples. Finally, shouldn't the experiments with a lower reproducibility be redone? This is of course now impossible to judge because we don't know the relationship between the correlations reported for reproducibility and those used later in the analysis.

Figure 1E shows has a much lower number of dots (significant changes) than indicated in Figure 1D (signature size). One explanation for this discrepancy is that here again two different measures of significance are being used. Furthermore, Figure 1E exemplifies why it is wrong to test reproducibility by ranking and analyze differences between samples using the values - the ranking in these two signatures will probably be the same, although we can't tell because a different correlation measure is being used.

Figure 1D is a correlation plot. Instead of using the heatmap of correlations a dendrogram would be much more appropriate and also more revealing. This looks like smoke and mirrors, also because a scale bar for the actual correlations is missing.

The legends for Figure 1 and for all the other figures are way to sparse to understand what is really being shown.

Figure 2C suffers from the same flaws as pointed out above.

Figure 2D requires a lot of improvement. The position of wildtype for each treatment is not visible. An explanation is needed for the ko's falling inbetween the IFNg and the FGF1 treatment and for the group below FGF1. The colour scheme is such that it is not clear which treatment the former samples received either. Fig 2D (and Fig 1D) would be much better to be judged if individual genes were shown.

Reviewer #2:

Summary:

The authors carryout a relatively large number of reverse genetic screens in human HAP1 cells. Shallow RNA sequencing data is used to measure transcript expression levels across ~12,000 genes and provides the cellular phenotype. First the cell transcript expression response to seventy stimuli is measured and the results indicate that the approach is able to capture some expected broad responses (e.g. ligands of the TGF beta superfamily) as well as reveal smaller novel signaling nuances (e.g. differences in expression ~ stimuli between Interferons beta, lambda and gamma). Next, 10 stimuli are selected and reverse genetic screens carried out using CRISPR/Cas9 for over 300 expressed but non-essential protein kinases. Expression data is generated from 64 HAP1 tyrosine kinase knock-out cell lines and evaluated in some detail. Knock-out specific signaling differences are observed and two specific examples presented, namely signaling differences between knock-out cell lines of the JAK and FGFR family members. Finally some small effect signaling differences are validated using qRT-PCR and the results are shown to be concordant with the expression data from the original screen.

General remarks:

This work provides a good proof of concept as a scalable approach for genotype to phenotype assessment in human cells. This study is restricted to human HAP1 cells (although the approach is likely applicable to different cell types) and the approach is restricted to cellular phenotypes based on transcriptional profiling. Overall good evidence is provided into the robustness of the approach as well as some insights into differences in stimuli response between members of two specific gene families. The key point of interest here is not so much the methods themselves rather the application

of this type of approach at scale in human cells. Thus this work is likely to be of broad interest to most genetic/biological researchers.

Major points

None.

Minor points

What is the relationship between signature size and signature robustness, the reproducibility of smaller expression levels is highly likely to be far less robust than for larger ones (10.1038/nmeth.2694).

Further details to support the estimated power to detect a >2 fold expression change should be provided, page 4 line 12 "we estimated that our depth range should enable us to call an expression fold change > 2 in more than two thirds of the transcriptome".

The GO analysis results for the ~200 signature genes in terms of pathways previously linked to the tested stimuli is a useful piece of information that has not been included (data not shown - page 4 line 29). These results should be included as, for example, a supplementary table.

The general description of Fig 1 D in the main text page 4 line 26 "stimuli produced signatures of up to ~200 genes, which displayed expected patterns" is not sufficient, please elaborate on how and why the observed patterns were expected.

If possible, quantifying the targeting efficiency using a numerical value, even if approximate, page 5 line 34 "targeting efficiency was high".

Page 6 line 34 has "is" at the end of the line that should be removed.

Figure 3 D has colors relating to the selected stimuli, including a key for the color to stimuli in the figure legend would be useful.

Reviewer #3:

In this manuscript the authors describe an approach suitable for parallel reverse genetics of mutant human cells by extensive low-coverage RNA-Seq. According to their findings, recovering as little as 1 million reads per sample, based on the approach they are able to differentiate differentially expressed genes induced by several different stimuli. The authors then use a panel of knock-out cell lines (kinases) and perturb the cells with a subset of the previously tested stimuli in order to generate a comprehensive view of the role of the tyrosine kinases in induced cell signaling. This is an interesting study with a potential to become very useful for performing reverse genetics in human cells. We have couple of issues that need to be addressed for this manuscript to be suitable for publication in MSB.

- 1-The authors use a haploid cell line, which should be easier to scale in terms of RNA count reads to see the differences in phenotypes. Can the authors simulate how to design experiments for diploid cells? There might be substantial differences between a haploid and a diploid cell line in terms of scalability, efficiency of knock-outs etc. If this method is presented as a general approach for parallel reverse genetics of human cells, it would be important to see at least one experiment done with a diploid cell line (which are more commonly used) to be able to compare the applicability of this method to any cell line based system. Otherwise, the method should be presented as haploid cell line-specific method.

- 2- We are unsure about some of the numbers in the manuscript. The authors state on page 3, line 19: "We also present a collection of over 300 human cell lines with knock-outs in non-essential kinases"

page 5, line 32: "We generated isogenic knock-out clones for over 300 expressed and non-essential protein kinases in HAP1 cells using CRISPR/Cas9".

We haven't found any experiment where these cell lines are used. The authors only use a subset (tyrosine kinases) of these knock-out cell lines which is in principle fine considering their interest in the cell signaling pathways. It is simply not necessary to emphasize so much the cell lines that were not used in the paper.

In addition: on page 7, line 21: "Besides demonstrating its suitability for studying cellular perturbations, we generated a proof-of-concept dataset of over 1800 samples in a collection of isogenic mutant cell lines"

Where is this number coming from?

3- The paper is too short to understand some of the methods. In particular we did not fully understand how they came up with stimulus response score? Also in the methods section, this part was not sufficiently explained. The figure legends could use more detailed information as well.

Minor points:

1- Figure 1C, based on this figure the authors claim that they can identify the vast majority of the expressed 12K genes with 1M reads. I am not sure if the figure supports this statement.

2- Figure 2A and Figure S2 is almost the same except the size difference. If the authors really want to show the kinome tree, they might want to indicate the name of the kinases on the FigureS2, where they have enough space.

3- Figure S1A describes the robustness of signatures but there is no statistical testing applied.

Reviewer #4:

In their manuscript entitled "Parallel reverse genetic screening in mutant human cells using transcriptomics" Nijman and colleagues describe a method for systematic reverse genetic screens in human cells using RNA sequencing as an output.

This is an interesting and important topic as methods for comprehensive phenotypic characterization of drugs and genetic backgrounds are still not widely available. Different approaches have been described in the past, including viability profiling, expression profiling on a limited gene set (e.g. ConnectivityMap) or microscopy, but it remains to be seen which method will be the most suitable for a broad range of application.

Having provided evidence for the overall satisfactory performance of their transcriptional profiling approach, the authors selected 10 stimuli to identify gene expression signatures to be used for functional annotation of small molecules. They validated their experimental platform in two different knock-out cell lines for which the impact of the gene on specific gene signatures has been anticipated. They could show that HIF1A knock-outs abolishes a gene signature dependent on hypoxia stimulation. Similarly, CTNNB1 knock-out cell lines abolish a signature derived from WNT3A stimulation.

They then go on to profile 64 isogenic knock-out cell lines, focusing to tyrosine kinases. They highlight the utility of their platform by focusing further on JAK1 knock-out and FGFR knock-out signatures. Their results show that JAK1 knock-outs render HAP1 cells insensitive to interferon stimulation, whereas this is not the case for the knock-outs of JAK2 and TYK2. For FGF1 signaling, they found that FGFR1 and FGFR3 knock-out cells abolish respective gene signatures whereas this is not the case for FGFR2 and FGFR4 knock-outs. They also show MINOR differences when FGFR1 knockout signature is compared to a FGFR3 knock-out signature.

The authors describe established a strategy for comparative transcriptional profiling of isogenic knock-out cells challenged with different small molecules. It is convincingly shown in the manuscript that the reverse genetics approach by shallow transcriptome can yield novel insight in genotype-phenotype associations.

The experimental approach described in this manuscript together with the dataset obtained in this study are of high interest to the scientific community and specifically to readers of Molecular Systems Biology. I would therefore support the publication of this manuscript in principle after revision.

Comments to be addressed:

1. The manuscript would gain from a better presentation of the results and clearer subdivision in the result section. Since this is not the first study using transcriptional readouts as a means to classify small molecules, the authors should also more broadly discuss advantages and disadvantages of their method compared to other published approaches.
2. The authors should describe in more detail the creation and characterization of the isogenic knock-out cell lines, including vectors used, sgRNA designs, selection strategy.
3. The authors should describe validation approach that the gene of interest was indeed removed?
4. Are knock-out cells associated with phenotypic alterations, e.g. proliferation etc.? A better description of the cells would be helpful.
5. Clearly specify if WNT3 or WNT3A was used. The text states WNT3A the figs state WNT3.
6. Fig 2D. Should the wt cells not be present in each of the response classes? I could only spot them in the "None" treated group.
7. Supplement Fig 3 and Page 6 line 16f: The authors note that their data suggest that various knock-out clones seemingly have an active hypoxia response under normoxia. This hypothesis could easily be tested, e.g. via HIF1 Western to further strengthen the reliability of their findings.
8. In Fig 2 E they show how different FGFR knockouts impairs FGF1-response signatures. They further discuss the subtle differences between effects of FGFR1 vs. 3 KO. In support they also highlight genes in Fig S6, however the respective gene names are hard to discern from the plots. It would be good to provide an alternative illustration so that readers can more easily identify those genes that are affected differently (e.g. a heatmap?)
9. In Fig 1E the authors show how CTNNB1 knockout affects WNT3A response signatures.

Given the central role of CTNNB1 in WNT3A induced signaling one would assume that all signature genes should be affected. However at least for 2 genes this is not the case. How is this explained?
10. To increase the accessibility of the results the authors could consider to provide their data in a browsable online format (comparable to e.g. the Cancer Therapeutics Response Portal; <http://www.broadinstitute.org/ctrp>).
11. It should be made sure that the data is deposited in public databases, processing code and scripts are provided and availability of cell lines is clarified.

1st Revision - authors' response

23 May 2016

Point by point response (original reviewers' comments are in *italic*). For clarity, we (re-)numbered the comments.

Reviewer #1:

1. *Summary and major points of critique*
The study by Gapp and co-workers is an expression-profiling analysis of a collection of CRSPR-Cas9 kinase knock-outs in human HAP1 (haploid) cells, under a set of 10 stimuli. This combination (the collection of knock-outs, the expression-profiling and the fact that the experiments have been carried out under a set of different conditions) would be expected to make an exciting study. However, as it stands, in essence this does not go beyond a methods paper and one whereby the methodology is not conceptually new. Furthermore, the paper lacks

a clearly new interesting biological finding (either general to all kinases or specific to a particular pathway), and certainly no interesting finding that is backed up by appropriate follow-up analyses/experiments. There are major issues with the way the data has been presented and the way that some of the steps/analyses have been carried out. Although the approach would seem to be exciting (based on previous studies in model organisms) this is not really exemplified in this study which altogether leaves an impression of superficiality, in particular because of the lack of an interesting new biological insight. Besides these major issues there are many minor issues regarding the presentation/write-up.

We appreciate the reviewer's point of view. However, the study serves mainly as a proof of principle study that demonstrates linking of genes to pathways using a combination of transcriptional profiling and engineered human cell lines. Although similar studies have indeed been performed in yeast it has not been demonstrated that this would also be applicable to more complex organisms. As such, this study is the first to demonstrate the scalability and sensitivity of this approach, even when investigating partially redundant or highly similar signalling pathways such as elicited by FGF or IFN, respectively.

2. *Individual comments*

In the introduction and in the main text ("first reverse genetics screens"), the authors state that reverse genetics has been limited to model organisms. This is incorrect for several reasons. Small-scale reverse genetics (one gene) in human cells has been going on for some time. Large-scale reverse genetics (many genes) has also been around for some time in the form of RNAi library screening approaches. There are obviously caveats to the latter that CRISPR-Cas9 will overcome, but such statements are incorrect. The (extremely short) introduction would benefit from a lot more subtlety.

We agree that the brevity of the introduction did not fully address these subtleties. We have reworked and expanded the introduction to address this point. We now also refer to reverse genetic studies using RNAi and its caveats with respect to CRISPR/Cas9.

3. *(With regard to the hurdles alluded to in the introduction the authors could add a third - cell type specific phenotyping.)*

We thank the reviewer for this suggestion and now also refer to cell type specificity in the introduction.

4. *The authors state that 300 knock-out clones have been generated for protein kinases with 90% of selected genes resulting in expandable clones. This superficiality is a pity and a missed opportunity. How many protein kinases are there in total? Did the authors target all? If not, what were the selection criteria? If they targeted all, then they should say something about essentiality (how many and which) discussing this in light of the systematically generated data of protein kinases from other organisms for which this is known (eg yeast). Furthermore, is the 90% a subset of the 300? If so how many exactly were made successfully (ie expandable: the 90% and 300 numbers come across as rounded-off numbers although they may be exact) and how many were then actually screened across the 10 conditions? It seems that although 300 kinase ko lines have been generated, only 64 have been profiled. This should be made more explicit (for example in the abstract) and the selection criteria needs to be described. It should also be noted that having only profiled a subset severely damages the otherwise potential systematic nature of this study. Initially doing all under a single condition and a subset under all 10 would already have been a step in the right direction.*

We apologize for any confusion regarding the selection of kinases due to the brevity of the text. There are 518 kinases in the human genome (Manning et al Science 2002). For each of these we know the ones that are expressed in HAP1 cells (Essletzbicher et al Genome Research 2014; this manuscript) and those that are required for proliferation under standard culture conditions (Blomen et al Science 2015). Filtering based on this information yielded approximately 300 kinases that are non-essential and expressed in HAP1 cells. Of these, 62 are tyrosine kinases and 55 were used in the screens because of availability. We have restructured and expanded this part of the text and now only mention those tyrosine kinases that we screened.

5. *The observation that some of the unstimulated ko's had a hypoxia-like response needs further work. Is this a result of the ko, or does it imply that a subset of unstimulated ko lines were inadvertently cultured under suboptimal conditions. If the former is true then this implies that these kinases all either have a role in suppressing such a response (probably unlikely if the number is large and includes several unrelated kinases). A third possibility is that this represents a frequently occurring indirect effect. Whatever the case the observation begs for a proper explanation.*

Following the reviewer's suggestion, we have further investigated the hypoxia-like response by investigating HIF1a levels and performing qRT-PCR. Besides selected clones that displayed the signature in the screen, we also generated new clones using an independent (second) gRNA. In summary, the levels of HIF1a correlated very well with the presence of the signature, providing a molecular explanation. Second, we noted that the signature was independent of the targeted genes. Thus, we conclude that this effect is due to a relatively frequent occurring indirect "passenger" event that is independent of the targeted gene.

6. *Given the technological nature of the current study it would be a good idea to describe in more detail the extent of the correlations between apparent response and sequencing depth and RNA concentration. This should lead to a precise recommendation on how this can be avoided in future studies. Although linear modeling can certainly help to correct for such effects, future studies would benefit from having to avoid such corrections.*

We thank the reviewer for this suggestion and now comment on this more directly in the text on page 7.

7. *Typo: page 6 last word "is" can be removed*

We removed this typo.

8. *Figures 1A, 2A, 2C are cartoon representations of the experiment set-up. These are a complete waste of space and should be removed to the supplemental figures, leaving more space for presenting actual findings. None of the experimental strategies being employed are at all difficult to comprehend by reading the text and it is difficult to understand why the authors think that it's a good idea to fill almost one third of their figures with such cartoons rather than data.*

Following the reviewer's recommendation, we have moved the cartoons to the supplement or adapted them to be more informative.

9. *All data being presented is derived through compilation of results on individual genes. While this is certainly a requirement for such studies, it would increase the confidence in the data if the authors also presented some of their results in the form of figures that show results on individual genes, for example with heatmaps. In this way things like similarity or differences can be better judged than through compiled results only.*

We appreciate this suggestion and now include several heatmaps that highlight responses of individual genes in Figure EV3.

10. *The authors present reproducibility in the form of a correlation of ranking (Spearman correlation Fig 1B, 2C). This is in itself already a red flag since it is much easier to get similar ranking than similar values. In addition although the correlation is derived by a test of similarity of ranking the example scatterplot doesn't seem to be plotting the rank, rather the expression level. This is incorrect. Even worse, when demonstrating differences a different measure is used, not based on ranking. It is wholly incorrect to first demonstrate reproducibility by one measure and then analyze differences by another. Reproducibility should be tested by the same type of correlation used to assess differences between different samples.*

We apologize if the figures caused any confusion but respectfully disagree that Spearman correlation is incorrect to assess reproducibility. The histogram shows the distribution of Spearman correlation coefficients across replicates, providing a sense of the reproducibility across many

samples. The inset shows a representative expression level scatter plot and removes the potential concern that the samples correlate by rank only, and not by expression values.

After the quality control step using correlations, our subsequent analyses are based on differential expression calling. This is not based on rank or correlations, but on effect sizes (fold changes), clearance margins (uncertainty intervals on expression levels, described in the Methods), and gene signatures sets. The use of gene signatures is a common technique to reduce dimensionality of the feature space and thus emphasize patterns in the subspace of interest.

11. *Finally, shouldn't the experiments with a lower reproducibility be redone? This is of course now impossible to judge because we don't know the relationship between the correlations reported for reproducibility and those used later in the analysis.*

We used several criteria to judge sample quality and excluded a small number of outliers for technical reasons ($15/1866 < 1\%$, because low read counts or gross inconsistency with prior profiles). Among the remaining samples, some replicate pairs are better correlated than others, but this is not crucial as our downstream analyses focus on gene sets that change by substantial fold changes. Changes in these signature genes, therefore, should by construction be detectable despite increased noise level among other genes. By focusing our analysis on gene sets, we could thus avoid repeating individual experiments.

Downstream analyses described in the methods are not based on the correlation values reported in the quality control steps.

12. *Figure 1E shows has a much lower number of dots (significant changes) than indicated in Figure 1D (signature size). One explanation for this discrepancy is that here again two different measures of significance are being used.*

We have not used different measures for calling significance and we apologize if the figures have caused confusion. The analysis methods used to determine signature genes are based on fold changes and uncertainty overlap criteria and are consistent throughout the manuscript (Methods section). Figure 1C indicates the signature size based on experiments performed in duplicate using 70 diverse stimuli. The data in Figure 1E is based on signatures that we recomputed after collecting additional replicates. As shown in Figure EV1, the size of the signatures tends to become smaller when more replicates (from multiple batches) are used. This explains the perceived discrepancy between signature sizes.

13. *Furthermore, Figure 1E exemplifies why it is wrong to test reproducibility by ranking and analyze differences between samples using the values - the ranking in these two signatures will probably be the same, although we can't tell because a different correlation measure is being used.*

See explanation above (point 10).

14. *Figure 1D is a correlation plot. Instead of using the heatmap of correlations a dendrogram would be much more appropriate and also more revealing. This looks like smoke and mirrors, also because a scale bar for the actual correlations is missing.*

We modified the figure with a monochrome heatmap and a scale bar. We also now provide an equivalent dendrogram representation in Figure EV1.

15. *The legends for Figure 1 and for all the other figures are way too sparse to understand what is really being shown.*

For most figure panels we have expanded the figure legends to address this matter.

16. *Figure 2C suffers from the same flaws as pointed out above.*

See explanation point 10.

17. *Figure 2D requires a lot of improvement. The position of wildtype for each treatment is not visible.*

As suggested, we now include wild type samples in the figure.

18. *An explanation is needed for the ko's falling inbetween the IFNg and the FGF1 treatment and for the group below FGF1.*

When using clustering methods like t-distributed stochastic neighbour embedding (tSNE), the precise patterns are different each time the algorithm is repeated. Thus, the position of the group of samples between IFNg and FGF will change when the procedure is repeated. This makes it hard to interpret the meaning of this observation. The only thing that can be concluded is that these samples are more similar to each other than to any of the other groups. We note that stochasticity in tSNE also explains why the updated clustering, which now contains wildtype samples, looks slightly different than in our previous submission. The layout cannot be guaranteed when updating the number of samples, even with a set seed for random number generation.

19. *The colour scheme is such that it is not clear which treatment the former samples received either.*

The dark blue samples concern those treated with resveratrol. These and some other samples cluster outside the dominant stimulus group partly because of imperfect clustering (algorithms do not guarantee global minima) and partly because of the less robust signal associated with small signatures. The intention of the tSNE clustering figure is merely to show that most cell lines respond as wild type to the stimuli, as expected. This clustering method is not well suited to call outliers. This is done using the violin plots as in Figure 3.

20. *Fig 2D (and Fig 1D) would be much better to be judged if individual genes were shown.*

We appreciate this suggestion and now include heatmaps with the individual genes in Figure EV3.

Reviewer #2:

21. *Summary:*

The authors carryout a relatively large number of reverse genetic screens in human HAP1 cells. Shallow RNA sequencing data is used to measure transcript expression levels across ~12,000 genes and provides the cellular phenotype. First the cell transcript expression response to seventy stimuli is measured and the results indicate that the approach is able to capture some expected broad responses (e.g. ligands of the TGF beta superfamily) as well as reveal smaller novel signaling nuances (e.g. differences in expression ~ stimuli between Interferons beta, lambda and gamma). Next, 10 stimuli are selected and reverse genetic screens carried out using CRISPR/Cas9 for over 300 expressed but non-essential protein kinases. Expression data is generated from 64 HAP1 tyrosine kinase knock-out cell lines and evaluated in some detail. Knock-out specific signaling differences are observed and two specific examples presented, namely signaling differences between knock-out cell lines of the JAK and FGFR family members. Finally some small effect signaling differences are validated using qRT-PCR and the results are shown to be concordant with the expression data from the original screen.

General remarks:

This work provides a good proof of concept as a scalable approach for genotype to phenotype assessment in human cells. This study is restricted to human HAP1 cells (although the approach is likely applicable to different cell types) and the approach is restricted to cellular phenotypes based on transcriptional profiling. Overall good evidence is provided into the robustness of the approach as well as some insights into differences in stimuli response between members of two specific gene families. The key point of interest here is not so much the methods themselves rather the application of this type of approach at scale in human cells. Thus this work is likely to be of broad interest to most genetic/biological researchers.

Major points

None.

We thank the reviewer for his/her positive feedback.

22. *Minor points*

What is the relationship between signature size and signature robustness, the reproducibility of smaller expression levels is highly likely to be far less robust than for larger ones (10.1038/nmeth.2694).

Indeed, larger signatures are expected to be more robust. Addressing this issue using the stimulus discovery dataset would give quite noisy results, as individual replicates can have considerable private gene expression profiles (Figure EV1B). However, the relationship becomes clear using the entire screen data. Appendix Fig S3 shows that responses to stimulation with IFN β and IFN γ (large signatures) achieve near-perfect overlap with our reference signatures. There is also a gradual decline in overlap among the stimuli with smaller signatures.

23. *Further details to support the estimated power to detect a >2 fold expression change should be provided, page 4 line 12 "we estimated that our depth range should enable us to call an expression fold change > 2 in more than two thirds of the transcriptome".*

We appreciate this suggestion and have extended the methods section to better explain the relation between sequencing depth and differential expression. We also include the power curves for 3 fold and 0.5 fold change to provide an improved sense of the relationship between sequencing depth and sensitivity (Figure 1B). The calculation leading to the figure is also provided as a vignette in our github package ExpCube.

24. *The GO analysis results for the ~200 signature genes in terms of pathways previously linked to the tested stimuli is a useful piece of information that has not been included (data not shown - page 4 line 29). These results should be included as, for example, a supplementary table.*

We now include the GO analysis in Table EV2.

25. *The general description of Fig 1 D in the main text page 4 line 26 "stimuli produced signatures of up to ~200 genes, which displayed expected patterns" is not sufficient, please elaborate on how and why the observed patterns were expected.*

We modified this sentence and instead added a table with GO enrichment results. In brief, gene enrichment shows that response genes are associated with the well-characterized stimuli. For example, one of the most enriched concepts upon BMP2 stimulation is a GO term referring to the BMP pathway.

26. *If possible, quantifying the targeting efficiency using a numerical value, even if approximate, page 5 line 34 "targeting efficiency was high".*

We adjusted the text to remove the ambiguity. In brief, edited clones were obtained in over 95% of the attempted cases.

27. *Page 6 line 34 has "is" at the end of the line that should be removed.*

This typo has now been removed.

28. *Figure 3 D has colors relating to the selected stimuli, including a key for the color to stimuli in the figure legend would be useful.*

We now include a colour key for this figure.

Reviewer #3:

29. *In this manuscript the authors describe an approach suitable for parallel reverse genetics of mutant human cells by extensive low-coverage RNA-Seq. According to their findings, recovering as little as 1 million reads per sample, based on the approach they are able to differentiate differentially expressed genes induced by several different stimuli. The authors then use a panel of knock-out cell lines (kinases) and perturb the cells with a subset of the previously tested stimuli in order to generate a comprehensive view of the role of the tyrosine kinases in induced cell signaling. This is an interesting study with a potential to become very useful for performing reverse genetics in human cells. We have couple of issues that need to be addressed for this manuscript to be suitable for publication in MSB.*

We thank the reviewer for his/her positive feedback.

30. *The authors use a haploid cell line, which should be easier to scale in terms of RNA count reads to see the differences in phenotypes. Can the authors simulate how to design experiments for diploid cells?*

The RNA sequencing part of the pipeline is identical for haploid and diploid cells. Haploid cells may express fewer absolute numbers of RNA molecules per cell but this has no impact as the method is based on total RNA.

31. *There might be substantial differences between a haploid and a diploid cell line in terms of scalability, efficiency of knock-outs etc. If this method is presented as a general approach for parallel reverse genetics of human cells, it would be important to see at least one experiment done with a diploid cell line (which are more commonly used) to be able to compare the applicability of this method to any cell line based system. Otherwise, the method should be presented as haploid cell line-specific method.*

There is an advantage in using haploid lines for genotyping of genome editing, which is simpler with a single allele. However, there is no fundamental technical limitation that would hamper the generation of collections of knockout cells in diploid cell lines. Indeed, many labs have made homozygous mutants in non-haploid cell lines. A direct comparison of the efficiency of making mutants in haploid vs. diploid cells would require a very substantial investment that we feel would not add to the study here. Therefore, we feel that even though we do not show an experiment in a diploid line, extending the concept to diploid cells is a reasonable extrapolation.

32. *We are unsure about some of the numbers in the manuscript. The authors state on page 3, line 19: "We also present a collection of over 300 human cell lines with knock-outs in non-essential kinases" page 5, line 32: "We generated isogenic knock-out clones for over 300 expressed and non-essential protein kinases in HAP1 cells using CRISPR/Cas9". We haven't found any experiment where these cell lines are used. The authors only use a subset (tyrosine kinases) of these knock-out cell lines which is in principle fine considering their interest in the cell signaling pathways. It is simply not necessary to emphasize so much the cell lines that were not used in the paper.*

We apologize for the confusion and for clarity now only mention the tyrosine kinases.

33. *In addition: on page 7, line 21: "Besides demonstrating its suitability for studying cellular perturbations, we generated a proof-of-concept dataset of over 1800 samples in a collection of isogenic mutant cell lines" Where is this number coming from?*

This number refers to the total number of RNA-seq samples that were generated. We have removed this number and only refer to number of clones and stimuli.

34. *The paper is too short to understand some of the methods. In particular we did not fully understand how they came up with stimulus response score? Also in the methods section, this*

part was not sufficiently explained. The figure legends could use more detailed information as well.

We have expanded the methods section and figure legends to clarify these matters. The stimulus response score is a descriptive term for a residual between observed points and our general linear models that correct signature overlap by technical covariates. We hope the methods are now easier to understand.

35. *Minor points:*

Figure 1C, based on this figure the authors claim that they can identify the vast majority of the expressed 12K genes with 1M reads. I am not sure if the figure supports this statement.

This statement refers to the top line in the figure. We adapted the colors to make the lines easier to distinguish. The top (black) line indicates that the number of expressed genes can be estimated consistently even at very low read depth. This is consistent with previous observations among the single-cell sequencing community. Here, we additionally provide computational estimates for ability to detect differential expression. We estimate that given our calling criteria and a sequencing depth of 2-4M reads, we could detect around 70% of the 12K genes as differential expression if they were to change by a factor of 2. That fraction rises to more than 90% if these genes were to change by a factor of 3.

36. *Figure 2A and Figure S2 is almost the same except the size difference. If the authors really want to show the kinome tree, they might want to indicate the name of the kinases on the FigureS2, where they have enough space.*

We have adapted this figure (now in Appendix Figure S1) and now focus on the tyrosine kinases.

37. *Figure S1A describes the robustness of signatures but there is no statistical testing applied.*

This figure illustrates the extent with which increasing replicate number improves the robustness of the signature. The distributions can be thought of as repeat calculations of signature size based on subsampled replicates. The distributions are thus the output of the testing procedure. From this plot, we conclude that the signature size changes dramatically from 1 replicate to 2 replicates, but much less thereafter. It is not appropriate to compute p-values between adjacent boxes on the boxplot as they are all based on the same set of 8 replicate samples.

Reviewer #4:

38. *In their manuscript entitled "Parallel reverse genetic screening in mutant human cells using transcriptomics" Nijman and colleagues describe a method for systematic reverse genetic screens in human cells using RNA sequencing as an output.*

This is an interesting and important topic as methods for comprehensive phenotypic characterization of drugs and genetic backgrounds are still not widely available. Different approaches have been described in the past, including viability profiling, expression profiling on a limited gene set (e.g. ConnectivityMap) or microscopy, but it remains to be seen which method will be the most suitable for a broad range of application.

Having provided evidence for the overall satisfactory performance of their transcriptional profiling approach, the authors selected 10 stimuli to identify gene expression signatures to be used for functional annotation of small molecules. They validated their experimental platform in two different knock-out cell lines for which the impact of the gene on specific gene signatures has been anticipated. They could show that HIF1A knock-outs abolishes a gene signature dependent on hypoxia stimulation. Similarly, CTNNB1 knock-out cell lines abolish a signature derived from WNT3A stimulation.

They then go on to profile 64 isogenic knock-out cell lines, focusing to tyrosine kinases. They highlight the utility of their platform by focusing further on JAK1 knock-out and FGFR knock-out signatures. Their results show that JAK1 knock-outs render HAP1 cells insensitive to

interferon stimulation, whereas this is not the case for the knock-outs of JAK2 and TYK2. For FGF1 signaling, they found that FGFR1 and FGFR3 knock-out cells abolish respective gene signatures whereas this is not the case for FGFR2 and FGFR4 knock-outs. They also show MINOR differences when FGFR1 knockout signature is compared to a FGFR3 knock-out signature.

The authors describe established a strategy for comparative transcriptional profiling of isogenic knock-out cells challenged with different small molecules. It is convincingly shown in the manuscript that the reverse genetics approach by shallow transcriptome can yield novel insight in genotype-phenotype associations.

The experimental approach described in this manuscript together with the dataset obtained in this study are of high interest to the scientific community and specifically to readers of Molecular Systems Biology. I would therefore support the publication of this manuscript in principle after revision.

We thank the reviewer for the positive feedback.

39. *Comments to be addressed:*

The manuscript would gain from a better presentation of the results and clearer subdivision in the result section. Since this is not the first study using transcriptional readouts as a means to classify small molecules, the authors should also more broadly discuss advantages and disadvantages of their method compared to other published approaches.

We have tried to better explain certain parts of the manuscript and have extended the figure legends to improve clarity and the introduction as well. However, we are limited by word count for this short format to fully unpack all information. We have also added some text to the discussion to place the study in a wider context. However, a comprehensive discussion of the pros and cons of the method would exceed the format.

40. *The authors should describe in more detail the creation and characterization of the isogenic knock-out cell lines, including vectors used, sgRNA designs, selection strategy.*

This is now included in the revised manuscript.

41. *The authors should describe validation approach that the gene of interest was indeed removed?*

This is now included in the revised manuscript. In short, all clones were validated by Sanger sequencing.

42. *Are knock-out cells associated with phenotypic alterations, e.g. proliferation etc.? A better description of the cells would be helpful.*

We comment on this issue in the text after introducing the KO cell lines. We have observed some phenotypic changes in a subset of the cell lines but have refrained from adding specific comments as these changes are difficult to quantify and interpret and were not the focus of this study (that deals with transcriptional changes). Overall, all cell lines were proliferating at similar speed.

43. *Clearly specify if WNT3 or WNT3A was used. The text states WNT3A the figs state WNT3.*

In all cases WNT3A was used. This is now corrected in the manuscript.

44. *Fig 2D. Should the wt cells not be present in each of the response classes? I could only spot them in the "None" treated group.*

We appreciate the suggestion and now include WT samples in Figure 2C.

45. *Supplement Fig 3 and Page 6 line 16f: The authors note that their data suggest that various knock-out clones seemingly have an active hypoxia response under normoxia. This hypothesis*

could easily be tested, e.g. via HIF1 Western to further strengthen the reliability of their findings.

Following the reviewer's suggestion, we have further investigated the hypoxia-like response by investigating HIF1a levels and performing qRT-PCR. Besides selected clones that displayed the signature in the screen, we also generated new clones using an independent (second) gRNA. In summary, the levels of HIF1a correlated very well with the presence of the signature, providing a molecular explanation. Second, we noted that the signature was independent of the targeted genes. Thus, we conclude that this effect is indeed due to a relatively frequent occurring indirect "passenger" event.

46. *In Fig 2 E they show how different FGFR knockouts impairs FGF1-response signatures. They further discuss the subtle differences between effects of FGFR1 vs. 3 KO. In support they also highlight genes in Fig S6, however the respective gene names are hard to discern from the plots. It would be good to provide an alternative illustration so that readers can more easily identify those genes that are affected differently (e.g. a heatmap?)*

We thank the reviewer for this suggestion and now include a heatmap for this experiment as part of Extended View Figure EV3.

47. *In Fig 1E the authors show how CTNNB1 knockout affects WNT3A response signatures. Given the central role of CTNNB1 in WNT3A induced signaling one would assume that all signature genes should be affected. However at least for 2 genes this is not the case. How is this explained?*

We agree that this is an interesting observation. The two genes are GAD1 and BHLHE22 and to the best of our knowledge have not been linked with WNT3A signaling, albeit they are consistently upregulated in our WNT3A stimulated samples. We can speculate that these genes might be peripheral markers of WNT3A response that is independent of beta catenin, but more in-depth work would be necessary to make firm conclusions. This type of finding highlights how reverse genetic screening with transcriptomics can be used as a hypothesis generation tool.

48. *To increase the accessibility of the results the authors could consider to provide their data in a browsable online format (comparable to e.g. the Cancer Therapeutics Response Portal; <http://www.broadinstitute.org/ctrp>).*

We certainly appreciate that a browsable format would be desirable. However, the development of such a data portal would require substantial resources that go beyond the ability of a single research group. The raw data, however, is available through the ENA sequencing archive. Processed expression profiles and analysis scripts are available as a download at zenodo.org.

49. *It should be made sure that the data is deposited in public databases, processing code and scripts are provided and availability of cell lines is clarified.*

The raw sequencing data is available at the ENA with project ERP012914. Several analysis tools are available on github in package ExpCube and scripts specific to this project are now available in a download at zenodo.org doi:10.5281/zenodo.51842 . Availability of cell lines is provided by Horizon Discovery.

Thank you for submitting revised manuscript to Molecular Systems Biology. We have now heard back from reviewer #3 who was asked to evaluate the revised study. As you will see below, this referee is satisfied with the modifications made. However, s/he lists two remaining concerns, which we would ask you to address in a revision. Both issues can be addressed by text modifications.

Since we are very flexible in terms of format, the Discussion can be extended as required in order to elaborate on the advantages/disadvantages and potential applications of the proposed approach (as also suggested by Reviewer #4 in the previous round of review).

REFeree COMMENTS

Reviewer #3:

While I am fine with the way most revisions were addressed I have two remaining comments that I feel are rather important:

(1) The fact that a haploid line was used in these experiments is not clearly mentioned neither in the main text nor the abstract, but needs to be mentioned there to clarify the approach. (The term "mutant" does not work here in lieu of 'haploid', neither in the abstract nor the main text. A haploid line is quite different from a regular human cell and it feels odd that this is nowhere clearly stated)

(2) In the Discussion section the authors state:

"In summary, we present an approach for parallel reverse genetics of mutant human cells based on shallow RNA-sequencing. Besides demonstrating its suitability for studying cellular perturbations, we generated a proof-of-concept dataset comprising 11 conditions in a collection of 64 isogenic mutant cell lines. This represents one of the largest transcriptomic experiments performed in a single cell line and demonstrates the scalability and suitability of the approach for exploring signaling mechanisms in human cells in a systematic manner".

As presently stated this paragraph appears as an overstatement and the authors need to be honest in describing their approach as a (presently) haploid cell line based method. Possible applications of their system in other cell types need to be discussed more comprehensively including the possible limitations of an approach in diploid cells.

2nd Revision - authors' response

06 July 2016

Point by point response (original reviewers' comments are in *italic*).

Reviewer #3:

50. *The fact that a haploid line was used in these experiments is not clearly mentioned neither in the main text nor the abstract, but needs to be mentioned there to clarify the approach. (The term "mutant" does not work here in lieu of 'haploid', neither in the abstract nor the main text. A haploid line is quite different from a regular human cell and it feels odd that this is nowhere clearly stated)*

We have edited the abstract (sentence starting "We conducted...") and introduction (last two sentences) to mention that we use haploid cells. We also mention here that we use the term "mutant" only to describe cell lines with knock-out genes, not ploidy.

51. *In the Discussion section the authors state:*

"In summary, we present an approach for parallel reverse genetics of mutant human cells based on shallow RNA-sequencing. Besides demonstrating its suitability for studying cellular perturbations, we generated a proof-of-concept dataset comprising 11 conditions in a collection of 64 isogenic mutant cell lines. This represents one of the largest transcriptomic experiments performed in a single cell line and demonstrates the scalability and suitability of the approach for exploring signaling mechanisms in human cells in a systematic manner".

As presently stated this paragraph appears as an overstatement and the authors need to be

honest in describing their approach as a (presently) haploid cell line based method. Possible applications of their system in other cell types need to be discussed more comprehensively including the possible limitations of an approach in diploid cells.

We extended the last discussion paragraph to mention transcriptional profiling in diploid cells. In short, we do not expect conceptual challenges in our methods for applications in diploid cells. We do concede, however, that generating mutants in a diploid parental cell line may be more difficult than in a haploid setting and that response signatures may also be affected. We would thus advise researchers to measure all relevant reference and control response signatures in a new cell system before quantifying differential transcriptional responses.

Corresponding Author Name: Jeff Gore
Journal Submitted to: Molecular Systems Biology
Manuscript Number: MSB-16-7033